# Stereotactic Radiotherapy for Hepatocellular Carcinoma, Radiosensitization Strategies and Radiation-Immunotherapy Combination

**DOI:** 10.3390/cancers13020192

**Published:** 2021-01-07

**Authors:** Luis A. Pérez-Romasanta, Elisabet González-Del Portillo, Ana Rodríguez-Gutiérrez, Ángela Matías-Pérez

**Affiliations:** Radiation Oncology Department, Hospital Universitario de Salamanca, Universidad de Salamanca, 37007 Salamanca, Spain; egportillo@saludcastillayleon.es (E.G.-D.P.); airodriguezg@saludcastillayleon.es (A.R.-G.); angelamatias@saludcastillayleon.es (Á.M.-P.)

**Keywords:** radiotherapy, hepatocellular cancer, immunotherapy, nanoparticles, protein kinase inhibitors

## Abstract

**Simple Summary:**

Radiotherapy is rapidly turning into a crucial component of multidisciplinary treatment for liver cancer because many patients are not surgical treatment candidates. Thanks to technical developments, radiotherapy have achieved high precision treatments, making it possible to eliminate tumor cells without severe damage to the liver and other organs. Stereotactic Body Radiation Therapy is an advanced radiotherapy technique able to eradicate malignant tumors wherever they are located in properly selected patients. The best use of radiotherapy, the most fruitful radiotherapy strategy, and the best way to combine it with other treatments for liver cancer are largely unknown. Radiosensitizers, agents that can potentiate radiotherapy, could broaden the radiotherapeutic landscape. Radiotherapy potentiation can be achieved with diverse treatments, not only drugs but also nanoparticles. In order to clear up the performance of radiotherapy in liver cancer management in the future and the best ways to potentiate its effects, considerable medical research is needed.

**Abstract:**

Stereotactic body radiotherapy (SBRT) is an emerging ablative modality for hepatocellular carcinoma (HCC). Most patients with HCC have advanced disease at the time of diagnosis, and therefore, are not candidates for definitive-intent therapies such as resection or transplantation. For this reason, various alternative local and regional therapies have been used to prevent disease progression, palliate symptoms, and delay liver failure. Stereotactic body radiation therapy is a non-invasive technique of delivering ablative doses of radiation to tumors while sparing normal or non-tumor hepatic tissue. Incorporation of SBRT in multidisciplinary HCC management is gradual, initially applied when other liver-directed therapies have failed or are contraindicated, and tried in combination with other locoregional or systemic therapies for more unfavorable conditions by more experienced teams. In order to improve SBRT therapeutic ratio, there has been much interest in augmenting the effect of radiation on tumors by combining it with chemotherapy, molecularly targeted therapeutics, nanoparticles, and immunotherapy. This review aims to synthesize available evidence to evaluate the clinical feasibility and efficacy of SBRT for HCC, and to explore novel radio-potentiation concepts by combining SBRT with novel therapeutics. It is expected that those approaches would result in improved therapeutic outcomes, even though many questions remain with regard to the optimal way to assemble treatments. Further trials are needed to evaluate and consolidate these promising therapies for HCC.

## 1. Introduction

Hepatocellular carcinoma (HCC) is the sixth most common cancer worldwide and the fourth major cause of cancer-related death [1]. Only a small fraction of patients is eligible for surgery. For the remaining patients, available therapies include stereotactic body radiation therapy (SBRT), percutaneous tumor ablation (PTA), intravascular therapy, such as trans-arterial chemo-embolization (TACE) or radio-embolization (TARE), and systemic therapy. Most patients require a combination of those therapies. Determining the best treatment options must consider the type, the stage, the patient’s performance status, co-morbidities, underlying liver function, and extra-hepatic disease [2].

At present, radiotherapy (RT) is turning into a crucial component of multidisciplinary treatment for HCC. It is mentioned as a plausible treatment alternative for early, intermediate, and advanced HCC patients in some guidelines, but kept as an ancillary alternative if other standard treatments are not feasible in other guidelines, owing to the lack of high-level evidence [3]. In some specific situations, such as inoperable patients with tumors situated near great vessels or the hilum of the liver, RT is commonly accepted as the preferred treatment option.

Radiotherapy designates a group of interventions based on the administration of high enough doses of radiation capable of killing tumor cells. Those treatments can be delivered from outside (external RT) or inside (internal RT) the body. Photon-based external RT is widely available and includes the highly precise short courses of SBRT, and the usually more protracted conventional RT. According the technology behind the control of the radiation beam fluence, the RT techniques are divided in two classes: unmodulated beam techniques, such as three-dimensional conformal RT (3D-RT) or dynamic conformal arc therapy; and modulated beam techniques, such as intensity modulated RT (IMRT) and volume-modulated arc therapy (VMAT). Particles, protons and carbon ions, are promising radiation types to treat patients with liver cancer, but they are not available for most centers. Internal RT with non-capsulated sources (metabolic treatments) or capsulated sources (brachytherapy) have also been used to treat liver cancer. A recent paper reviewed the RT armamentarium for different stages of liver cancer [4]. The technological description of all those techniques is beyond the scope of this review.

Provided the encouraging early results with SBRT for patients with early and advanced HCC, there is growing interest to explore the use of SBRT alone and in combination with other local and systemic therapies to optimize tumor responses and improve survival by maximizing local control and metastatic dissemination avoidance. Recent data have suggested that RT may act synergistically with immunotherapy to stimulate antitumor immune responses, adding new possibilities to RT indications. This review aims to analyze the effectiveness and safety of SBRT, and current radio-potentiation strategies integrating SBRT in combined treatments for HCC.

## 2. Hepatocellular Carcinoma Radiobiology

A common assertion regards HCC as a radioresistant tumor. Radiosensitivity is a measure of the surviving fraction of clonogenes after a given radiation dose. A frequently used parameter to describe radiosensitivity is the fraction surviving at 2.0 Gy single dose (SF2). As the relationship between radiation dose and surviving fraction is non-linear, diverse mathematical formulations have been proposed to fit the radiation survival curve. The linear-quadratic (LQ) model has become the most popular calculator to analyze and predict responses to ionizing radiation in the laboratory and in the clinic. The LQ model provides a simple equation between cell survival and delivered dose: S = exp (−αD−βD^2^). The LQ model parameters are α, β, and α/β-values. Several authors have described radiosensitivity model parameters, SF2 and LQ, for HCC cell lines in clonogenic assays [5,6,7,8]. Table 1 summarizes the most relevant radiosensitivity parameters for HCC.

The HCC radiosensitivity spectrum fits in the range of common non-HCC human tumors treated with RT [9]. Liu et al. [10] have found a not-significant difference between SF2 values for HCC from 58 samples with SF2 values for all human tumor types [7,9,10]. Therefore, there is no justification for the common statement that HCC is a radioresistant tumor.

Yeoh et al. [11] confirmed recently that HCC cell lines have a wide spectrum of radiosensitivity values. They obtained 16 patient-derived HCC lines, exposed organoid cultures to 8 Gy, and quantified the survival fraction, classifying them as sensitive, moderately sensitive, and resistant to RT. The radiosensitivity of the cell lines were distributed mainly into moderately sensitive (43.75%) and resistant (37.5%) groups, but approximately one-fifth of the cell lines (18.75%) were classified as sensitive [11].

## 3. Clinical Studies of SBRT for HCC

Since radiobiology is not an impediment for the curative use of radiation on HCC, technology should not be either whenever the treatment fulfills high precision requirements. High precision is characteristic of SBRT. In the last two decades, many studies have assessed SBRT safety and efficacy in primary liver tumors [4,12,13]. Data revealed high local control rates and low rates of severe toxicity. Besides safety and efficacy considerations, other rationales for the typical short courses of SBRT are that they do not significantly interrupt systemic treatments and they are less invasive and disturbing for the patient than other types of liver directed therapies.

### 3.1. Initial Developments

The first publication of SBRT for liver tumors included 10 patients reporting an objective response rate of 70% [14]. Several prospective dose finding trials were conducted afterwards (Table 2). Méndez Romero et al. obtained local control (LC) in six out of eight HCC patients after applying two different dose fractionation schedules: 37.5 Gy in three fractions and 25 Gy in five fractions [15]. Two episodes of Radiation-Induced Liver Disease (RILD) and one death due to hepatic failure was observed. No late toxicity was found. Tse et al. included 41 patients with relatively large unresectable tumors [16]. After SBRT (median dose of 36 Gy in six fractions), the median survival time was 13.4 months with 1-year infield LC rates of 65%. Within the first 3 months after treatment, no RILD grade 4 or treatment-related death was noted. The promising results from previous trials established the ground for further phase II studies of three to six-fraction SBRT.

Cárdenes et al. [17] performed a phase I trial using 36 to 48 Gy in three fractions for Child-Turcotte-Pugh (CTP) Class A and 40 Gy in five fractions for CTP Class B patients with primary HCC and reported 100% LC rate at a median follow-up of 24 months with a 2-year survival of 60%. In an update of their combined phase I/phase II trial, the authors reported a 2-year LC of 90% and Overall Survival (OS) of 67%, with the majority of the patients receiving 48 Gy/3 fractions for CTP Class A and 40 Gy/5 fractions for CTP Class B patients [17]. A phase II trial from Toronto included patients with CTP Class A disease and used doses ranging from 24 to 54 Gy in six fractions over locally-advanced tumors in 102 patients [18]. The results revealed a 1-year LC rate of 87% and a median OS of 17 months. Grade 3 or higher toxicities were noted in 36% of patients mainly due to asymptomatic lab abnormalities. Death was possibly related to treatment in seven patients.

These initial trials concluded that SBRT was feasible, with acceptable toxicity for CTP Class A patients, and effective with encouraging local control, providing strong foundations for studying SBRT for HCC in randomized trials. Optimal fractionation and technological refinements, particularly for functionally compromised patients, have to be found.

### 3.2. SBRT for Small Tumors

There have been several studies, mostly retrospective, of patients with HCC treated with SBRT in diversified indications: small tumors, large tumors, multifocal disease, tumors with vascular invasion, patients on the waiting list for a liver transplant (LT), and even selected metastatic cases. As obviously expected, the best results have been reported for small tumors (Table 3).

One of the largest unicentric studies included 185 liver tumors smaller than 5 cm in size and used SBRT in five fractions, reaching doses up to 40 Gy, which could be considered moderate doses, obtaining three-year LC and OS rates of 91% and 70%, respectively [19]. Mathew et al. [20] reported long-term outcomes for HCC patients without tumor vascular invasion, who were unsuitable for other local treatments from two North American centers. The pooled survival results on 297 patients with 436 lesions showed a 3-year OS rate of 39.0%. Better prognosis was related with LT after SBRT, CTP Class A, alpha-fetoprotein < 10 ng/mL, and ECOG performance status 0. Cumulative local recurrence was 13.3% at three years, significantly lower for smaller tumors and in patients able to collaborate with breath-hold techniques. Worsening of CTP score by more than two points three months after SBRT was seen in 15.9% [20].

As there is currently a limited number of studies with long-term outcomes after SBRT for small HCC, pooled analysis and systematic reviews provide the best source of information. Sixteen studies (973 patients with 1034 lesions), 14 of them retrospective, were included in a systematic review with qualitative and quantitative analyses [21]. Average tumor diameter was 23 mm and most patients were in good general condition. Only 22.9% of patients were treatment naive. Mean weighted 3-year LC and OS rates were 93% and 73.4%, respectively. Toxicity reached grade 3 or higher in only 5.3% of patients, without treatment-associated mortality. The authors considered that the quality of findings on toxicities was not optimal due to deficient reporting and the retrospective nature of the studies.

Another systematic review included 32 studies involving 1950 HCC patients [22]. The median value of median tumor sizes among studies was 3.3 cm. Pooled 3-year LC rate was 83.9%, and 3-year OS rates was 48.3%. Tumor size, but not radiation dose, showed a significant impact on OS and a marginal difference for the 1-year LC rate. Pooled rates of hepatic and gastrointestinal grade ≥3 complications were 4.7% and 3.9%, respectively, significantly correlated with the CTP class.

Those studies confirmed the feasibility and efficacy of SBRT on small tumors, offering high LC probability with OS apparently comparable with PTA and surgery. For patients with poor liver function but small tumors, SBRT proved to be safe and effective in a prospective study, providing high probability of local control with mild toxicity [23]. To establish the place of SBRT in the treatment of small HCC, randomized trials should compare it face to face with hepatectomy or PTA.

### 3.3. SBRT Versus Resection

Liver resection is the standard of care therapy for operable patients with early-stage HCC. Three Chinese groups have conducted retrospective studies to compare SBRT and hepatectomy (Table 4), and a Chinese randomized trial (ClinicalTrials.gov Identifier: NCT03609151) confronting resection and SBRT is currently recruiting patients.

Su et al. [24] compared SBRT (*n* = 82) to resection (*n* = 35) in patients with CTP-A disease and one or two small nodules, reporting comparable survival, before and after propensity score matching, with fewer complications, such as hemorrhage, pain, and weight loss, in the SBRT group, despite more adverse conditions in that group of patients. The percentage of older patients was higher in the SBRT group, some of them medically unfit for surgery. [24]. Contrarily, Nakano et al. reported superior survival in patients with CTP-A/B disease and one to three small HCC tumors treated with hepatectomy (*n* = 254) compared to SBRT (*n* = 27). Propensity score analysis revealed that among 54 patients with surgical resection and 27 with SBRT, the 5-year OS and DFS rates were 75.2% vs. 47.8% (*p* = 0.0149) and 33.8% vs. 16.4% (*p* = 0.0512), respectively [25]. Given those contradictory results, it is not clear that SBRT could be recommended as alternative treatment for operable patients with early HCC.

The most recent and probably most meaningful study to compare surgery and SBRT included 317 naïve small HCC patients with hepatitis virus-related cirrhosis treated with surgery (*n* = 195) or SBRT (*n* = 122) followed for five years. After proper selection, 104 patients from each group were matched for further analysis. No significant difference was found between the two groups in the OS and PFS rates: the 5-year OS rate was 70.7% in the resection group and 71.0% in the SBRT group; the 5-year PFS rate was 47.3% in the resection group and 49% in the SBRT group. No deaths were observed after resection or SBRT [26]. At least for the treatment of naïve virus-related cirrhotic patients, SBRT is a potential effective alternative to hepatic resection.

### 3.4. SBRT Alone for Large Tumors or with Vascular Tumor Thrombosis

The risk of vascular invasion and dissemination increases with size. Hypofractionated RT is occasionally offered as a salvage treatment for recurrent HCC with vascular invasion. Tumor thrombus response is commonly seen and median survival time close to one year could be expected [27]. Comparison between SBRT and conventional RT in retrospective series of patients with HCC with portal vein tumor thrombosis (PVTT) favors SBRT in term of response, local control, and survival [28], but the results have been inferior compared with patients without vascular invasion [28,29]. Furthermore, the optimal dose schedule to follow in order to reduce hepatic toxicity in centrally located tumors is unknown. Due to poor results, combination treatments including SBRT have been proposed supported by retrospective analysis of different treatment options.

For instance, a network meta-analysis assessed the outcomes for eight regimens applied on advanced HCC with PVTT: SBRT alone, SBRT combined with TACE, conventional RT combined with hepatic arterial infusion chemotherapy (HAIC) or TACE, TACE plus sorafenib, HAIC alone, sorafenib alone, and TACE alone. Fifteen studies involving a total of 2359 patients were analyzed. The pooled results showed that conventional RT plus HAIC was the best strategy in treating advanced HCC with PVTT, followed by RT plus TACE [30].

Large tumors, especially those with vascular invasion, are not the best candidates for RT-alone strategies but initial treatment with SBRT for those patients unsuitable for surgical or intravascular procedures can achieve thrombus shrinkage and flow recovery in the portal vein. Therefore, SBRT upfront could offer the patients a chance to receive additional treatment.

### 3.5. SBRT as a Bridge to Transplant

Although SBRT is a local therapy most frequently used for patients unsuitable for surgery or LT, it can also be used as a bridge procedure to downstage lesions that do not meet the Milan or San Francisco criteria for LT or to prevent tumor progression while patients are on a waiting list for LT. In 2011 O’Connor et al. evaluated a series of 10 patients with 11 HCC lesions treated with SBRT while on the LT waiting list. Local control over this period was 100% and all patients underwent LT without increased surgical complications. The pathological exam revealed a complete response in 3 of the 11 tumors (27%) [31]. Additional reports have confirmed the efficacy of SBRT in this setting [18,32,33,34].

The most common bridging treatment before LT is TACE, but it has not been proven yet that TACE is the best strategy. Sapisochin et al. analyzed retrospectively 379 patients treated with SBRT (*n* = 36), PTA (*n* = 244) and TACE (*n* = 99), resulting in similar effectiveness both in terms of survival from the time of listing (*p* = 0.7) and from the time of transplant (*p* = 0.4). [35]. In a randomized phase II trial, 54 patients who were planned for LT were randomized to either ambulatory SBRT (*n* = 24) or two TACE treatments (*n* = 30). Patients treated with SBRT received a median total dose of 45 Gy delivered over five fractions. SBRT appeared as effective as TACE at controlling HCC prior to LT, engendered less toxicity, and eliminated the need for hospitalization [36]. Therefore, SBRT is safe and effective as a bridge to LT and can be offered as an option alongside other bridging therapies.

### 3.6. SBRT Versus Percutaneous Tumor Ablation (PTA)

Both SBRT and PTA are effective liver directed treatment options for HCC, but whether SBRT offers comparable clinical results as PTA is still questionable. The reports of single hospital series support the assumption of parallel outcomes between the two treatment modalities, but retrospective series with limited number of patients are not able to give a definitive answer [37,38]. Multicentric studies and metanalysis provide the best available evidence at the moment (Table 5).

Kim et al. first compared SBRT to PTA in a single institution, and later in a multicenter study [39,40]. The multicentric study cohort involved 2064 patients, 76% treated with PTA and 24% treated with SBRT. A high proportion of patients (56.5%) had failed previous treatments. The SBRT group had unfavorable prognostic factors compared to the PTA group, including BCLC stage, tumor size, and frequent history of local treatment. Local recurrence rate at 3 years was higher in the PTA group (27.9% versus 21.2%; *p* < 0.001). Propensity score-matched analysis confirmed the lower risk of local recurrence after SBRT (HR 0.36, *p* < 0.001). The benefit of SBRT over PTA was associated with tumor size (>3 cm), subphrenic location, and recurrence after TACE.

Ten retrospective studies with 2732 patients have been included in a metanalysis to compare treatment outcomes for SBRT and PTA. SBRT demonstrated significantly higher 1- and 3-year LC than PTA, but no significant difference between both groups for OS. For those patients who needed a bridge therapy, no differences in transplant rate and pathological necrosis rate in the explanted livers were found [41]. Another metanalysis included eight studies involving 1990 HCC patients. Local control rates at 2 years were similar after SBRT or PTA (84.5 vs. 79.5%; *p* = 0.431) but the OS analysis favored PTA (Odds ratio = 1.43; *p* = 0.023) [42]. The notion of better survival after PTA than after SBRT has been challenged in a national cohort of early staged HCC patients, where PTA and SBRT resulted in no significant difference in survival, 90-day hospitalization, or costs [43]. Contrarily, using the National Cancer Database (USA), Rajyaguru et al. [44] carried a propensity score matched analysis and found better 5-year OS after PTA (29.8%) than after SBRT (19.3%) on 3684 and 296 nonsurgically managed patients with stage I or II HCC [44].

In the absence of randomized controlled trials comparing survival after SBRT and PTA, selections bias cannot be excluded to explain the observed differences in survival. As a virtual comparison exercise, Seo et al. [45] conducted a simulated randomized trial using a Markov model analysis to compare the projected survival for patients with small (≤3 cm) tumors after treatment with PTA or SBRT. The model parameters were obtained by a systematic review of the literature. According to this model, the expected median survival of SBRT and PTA are almost identical (76.5 months vs. 77.4 months), with some advantage for SBRT in tumors 2 cm and larger [45].

Overall, these data support the use of SBRT as primary treatment for inoperable HCC, but do not displace PTA as first option for suitable patients. The diverse treatments proposed for inoperable HCC patients should be compared face to face in randomized clinical trials to establish the first-line local therapy. At least, three such studies are recruiting patients at the moment in small tumors (ClinicalTrials.gov Identifier: NCT03898921), recurrent tumors (ClinicalTrials.gov Identifier: NCT04047173), and perivascular tumors (ClinicalTrials.gov Identifier: NCT04434989).

### 3.7. SBRT Versus TACE

Transarterial chemoembolization (TACE) has become the mainstay of treatment for unresectable HCC and for PTA-ineligible patients due to poor liver function and/or difficult location. Few studies have compared TACE and SBRT to date (Table 6).

A large single-institution study involved 209 patients with 287 small HCC tumors (2.3–2.9 cm) treated with TACE or SBRT [46]. Despite 2-year LC rates being higher with SBRT than with TACE (91% versus 23%), no OS differences were noted. Patients treated with TACE were more prone to develop grade 3+ toxicity than patients treated with SBRT (13% versus 8%). For a little more advanced tumors, with sizes between 3 and 8 cm, Shen et al. obtained a 3-year LC rate of 73.3% for the SBRT group and 63.0% for the TACE group in a retrospective single institution study with 188 tumors. Independent predictors for LC were treatment modality, gender, and recurrence status. After propensity score matching analysis, the differences in 3-year LC became larger in favor of the SBRT group (77.5 vs. 55.6%; *p* = 0.007). Patients treated with SBRT had higher survival chances than those in the TACE group (3-year OS = 55.0 vs. 13.0%; *p* < 0.001). Only in the subgroup of newly diagnosed tumors, there was no difference in local control and OS between both treatments [47]. Local control, intra-hepatic control, and PFS reached more benefit from SBRT compared to TACE in a multi-institutional retrospective study on 326 patients with inoperable BCLC-A stage HCC. Using multivariate analysis, treatment modality was a significant covariate associated with local (HR = 1.59; *p* = 0.04) and intrahepatic control (HR = 1.61; *p* = 0.009), but not with survival [48].

Certainly, controlled trials are needed in order to avoid the selection bias implicit in retrospective studies. The TRENDY trial (ClinicalTrials.gov Identifier: NCT02470533) is a multicenter trial that will compare TACE with SBRT in treatment-naïve patients with HCC ineligible for surgery or PTA. The Instituto Clinico Humanitas has just completed recruitment for a randomized trial (ClinicalTrials.gov Identifier: NCT02323360) of salvage SBRT after incomplete TACE versus rechallenge with TACE for inoperable patients. A Chinese trial (ClinicalTrials.gov Identifier: NCT03326375), with similar design, is still recruiting patients.

Therefore, SBRT is finding a place for early or moderately advanced tumors in inoperable patients unsuitable for PTA as definitive treatment or as bridge to transplant. In this setting, SBRT convenes a noninvasive nature with efficacy, at least in the same amount as TACE. NCCN guidelines listed RT since 2016 as a suitable option for the treatment of advanced HCC but current NCCN Guidelines Version 3.2020 consider SBRT in both resectable and nonresectable cases.

### 3.8. SBRT Combined with TACE

#### 3.8.1. SBRT Combined with TACE for Small Tumors

Tumor tissues are commonly not completely eliminated through TACE. Approximately only 22–50% of the tumor is destroyed in histopathological necrosis evaluations. Cooperation between TACE and SBRT could improve the rate of complete necrosis after combined treatment, as has been suggested by a study by Honda et al., where SBRT (48 Gy per 4 fractions) was performed no later than two months after TACE in 68 patients with small (≤3 cm), solitary, and hypervascular HCC [49]. Complete response rates were significantly greater for SBRT than for TACE (96.3% vs. 3.3%; *p* < 0.001). There were no induced fatalities after the combined treatment. For naïve patients, the combined therapy obtained better 3-year OS than TACE alone group (100% vs 66.1%). Median DFS time was 15.2 months in the combination therapy group but only 4.2 months in the TACE alone group. Jun et al. [50] compared retrospectively 85 patients who underwent combined treatment and 114 patients who underwent TACE alone for tumors measuring less than 5 cm. Rate of LC, PFS, and OS were compared after propensity-score matching. The combined treatment group showed significantly higher 1- and 3-year LCR and PFS than the TACE alone group. The benefit in LC and PFS was restricted to patients with fewer than three tumors. However, OS was not different between treatment groups [50].

It is not clear if small tumors need TACE in addition to SBRT. Takeda et al. prospectively analyzed outcomes of patients who received SBRT (35 to 40 Gy in five fractions) and optional TACE for small (≤4 cm) solitary HCC. Ninety evaluable patients were followed for a median of 41.7 months after treatment. The group was a mixture of treatment-naïve and residual or recurrent tumors. Though 32 patients did not receive TACE, 48 received insufficient TACE, and only 10 attained complete embolization, almost all patients achieved local control. Overall survival and liver cause-specific survival were not different between treatment groups [51]. Kimura et al. reported on 150 patients with 185 small lesions treated with SBRT alone (*n* = 28) or TACE plus SBRT (*n* = 122). The 2-year overall, progression-free, and local progression-free survival times in SBRT alone and combination groups were similar [52]. Analogous conclusions can be drawn from European series [53].

Based on the fact that moderate radiation doses induced changes in tumor permeability and perfusion in preclinical studies, a strategy of SBRT preceding TACE for the purpose of enhancing TACE delivery and efficacy has been tested in a small pilot study, suggesting an opportunity for TACE sensitization by radiation. Single-dose SBRT (7.5 or 10 Gy) followed by TACE within 24 h was feasible and tolerable. Perfusion MRI revealed acute changes in tumor permeability/perfusion after SBRT as expected [54].

In summary, SBRT after TACE for small tumors improves the rate of complete necrosis, local control, and DFS, with no apparent effect on OS. Combined treatment is recommended for subgroups of patients with no previous treatments and fewer than three tumors. The need of combined treatment is dubious if SBRT can be used.

#### 3.8.2. SBRT Combined with TACE for Large Tumors

Large HCC often corresponds to the BCLC B (intermediate) and C (advanced) stages, most commonly treated with TACE or sorafenib, respectively. However, TACE rarely eradicates large HCCs and renders a 5-year OS rate of less than 10%. Furthermore, there is little effect of sorafenib in patients with macrovascular invasion and/or metastasis. As treatment results are suboptimal for advanced stages, new strategies must be evaluated. A recent meta-analysis has looked at the results of the combination of TACE with different RT modalities, but not SBRT, concluding that TACE followed by conventional RT may be associated with better survival and increased tumor response [55].

There are at least two hypothetical advantages of combining TACE and SBRT for large tumors [56]: spatial cooperation, as TACE is less effective at the periphery of the tumor while SBRT yields more tumor cell killing in the well oxygenated tumor areas, probably located in the tumor margins; and synergism, because the cytotoxic agents used in TACE may produce tumor radio-sensitization and a more pronounced tumor response. Several studies supported the combined treatment strategy. Kang et al. [57] reported a study in which 47 patients with large (up to 10 cm) inoperable tumors and PVTT in a quarter of cases, achieving 76.6% overall response rate to SBRT following one or multiple TACE procedures. The 2-year LC, OS, and PFS rates were 94.6%, 68.7%, and 33.8%. This seminal study highlighted the promising results that could be achieved with combined treatment. Other authors have compared retrospectively the outcomes of patients who received SBRT followed by TAE or TACE (*n* = 77) with patients who received SBRT alone (*n* = 50) for tumors larger than 5 cm [58]. Overall survival and distant metastasis-free survival rates were significantly higher in the combined treatment group, but PFS and survival without local relapse were similar between the two groups. Even in patients with unresectable huge HCC (>10 cm), the combined treatment showed to be a safe and effective option, reporting an objective response rate of 79.1% and a median survival time of 12.2 months [59].

Zhao et al. [60] conducted a meta-analysis to test the hypothesis of higher efficacy of combined treatment versus SBRT alone for unresectable tumors. The results, on 980 patients, showed that the combined treatment group had longer OS (*p* = 0.0002), higher complete response rate (*p* = 0.01), and a higher disease control rate (*p* = 0.02) than the SBRT alone group, while no significant difference in total adverse events were found between the two groups. The combined therapy benefit was not proven for patients with PVTT.

Therefore, SBRT + TACE appears to be more effective than SBRT alone in treating unresectable HCC, but the advantage seems to be restricted to the patients without vascular invasion.

#### 3.8.3. SBRT Combined with TACE for Tumors with Vascular Invasion

The recommended treatment for HCC patients with vascular invasion is the use of a systemic TKIs (Tyrosin-Kinase Inhibitors). It is questionable that RT, alone or combined with locoregional therapies, may be helpful in this patient group. A randomized phase II study reported by Yoon et al., including treatment-naïve patients with good functional status, compared sorafenib versus TACE combined with hypofractionated RT (45 Gy in 2.5–3.0 Gy per fraction) [61]. All patients had macroscopic vascular invasion, and most patients had multifocal disease and large tumors. Though sorafenib is considered the standard of care, patients in the combined TACE-RT group had an improved PFS at 12 weeks (86.7% vs. 34.3%, *p* < 0.001) and a longer median OS (55 weeks compared to 43 weeks; *p* = 0.04). All the patients in the combination treatment group received the planned treatment without toxicity impediments.

The previous experience is not unique, and a growing number of centers recommend SBRT for unresectable HCC with PVTT with the aim to shrink the tumor thrombus, preserve portal venous flow, and facilitate consecutive intravascular or surgical treatments. That strategy can achieve median survival rates in the order of one year, longer for those with indication for TACE after SBRT [62,63].

The favorable survival outcomes coming from the combined therapy support the notion that SBRT and TACE are complementary and cooperate to improve LC rates also in patients with PVTT. Multiple ongoing clinical trials are currently evaluating the combined strategy (ClinicalTrials.gov Identifier: NCT01918683, NCT02507765, NCT02513199, and NCT02794337) or comparing SBRT to TACE (ClinicalTrials.gov Identifier: NCT02182687 and NCT02470533).

### 3.9. SBRT Combined with TARE

Few studies have explored toxicities and outcomes of patients who were treated with SBRT after transarterial radioembolization (TARE). There is a concern for radiation toxicity with summation of the radiation dose coming from TARE and SBRT. Indeed, the liver tolerance to TARE is reduced by previous external beam RT [64]. Hardy-Abeloos et al. [65] reported the results on 99 patients treated with SBRT after TARE (*n* = 31) or TACE (*n* = 68). Severe toxicity from SBRT post-TACE was not significantly different as toxicity from SBRT post-TARE. Tumor control and survival were also similar. Consequently, TARE and SBRT can be combined with acceptable tolerability, at least for tumor volumes smaller than 300 cc.

### 3.10. SBRT for Extrahepatic Metastasis

The increased survival of patients with advanced HCC is probably behind the increased number of extrahepatic metastasis observed in recent years. Metastatic dissemination from HCC involves most frequently the lung, but every organ can be affected. Recently, SBRT and other high-precision RT techniques have been used as a palliative treatment to reduce symptoms and complication risks derived from metastatic implants in HCC patients [12,66,67,68,69,70]. Surgery has been already used in oligometastatic disease, particularly in selected patients with limited number of lymph node metastases. As in other malignancies, there is a growing interest in non-surgical options for the oligometastatic patient. The role of SBRT as a radical non-invasive treatment option in this setting is promising [69,70]. Even in patients with extrahepatic metastasis, the achievement of primary tumor local control with SBRT can promote longer survival compared to patients treated with sorafenib alone [71].

## 4. Radiosensitization Strategies

Radiosensitizers are substances or conditions capable of increasing the sensitivity of tumor cells to radiation, overcoming their radio-resistance. Many substances have been developed as HCC radiosensitizers: chemotherapy, targeted therapies, and nanoparticles. Though SBRT clinical effects are promising, it is worthwhile to review the different approaches to increase SBRT potency by combination with radiosensitizers.

### 4.1. Chemotherapy

The potentiation of RT with chemotherapy have been exploited up to maximal tolerance in different tumor types, but not for HCC, where this strategy has been adopted infrequently. Hypofractionated or conventionally fractionated RT has been occasionally associated with chemotherapy radiosensitizers such as 5-Fluorouracil, Doxorubicin, Thalidomide, and Capecitabine in advanced HCC with modest results [72,73,74,75,76,77]. Concomitant systemic chemotherapy, intravenous or intra-arterially infused, have not been clinically tested in combination with SBRT for HCC patients.

Since traditional chemotherapy agents are not expensive therapies, the radiosensitization strategy combining those agents and SBRT warrants basic and clinical investigation considering the high incidence of HCC in some low-income countries. The experience with Vinorelbine is a good example. Recently, vinorelbine potentiation effects on 16 HCC patient-derived xenograft models was determined in vitro. The combined treatment significantly increased the antitumor effect through diverse mechanisms compared with RT or Vinorelbine treatment alone [11].

### 4.2. Sorafenib

Sorafenib (SOR) is an oral multikinase inhibitor accepted as a first line treatment for patients with newly-diagnosed advanced HCC, since it is able to improve survival, although the improvement is modest. More recently, Lenvatinib, another multikinase inhibitor, has joined SOR as treatment option for that situation. New drugs, Regorafenib and Cabozantinib, have been approved as second-line treatments. Given the promising results with SBRT, especially combined with TACE, for patients with advanced HCC, it seems reasonable to combine effective local treatments with systemic therapies in order to improve tumor control. Synergic treatment of SBRT and target agents is an ongoing research field, both in the laboratory and in the clinic [78,79].

#### 4.2.1. Preclinical Studies

Sorafenib and RT sequencing could be crucial in order to maximize the interaction between them, but experimental results have been confounding. Li et al. showed that SOR, 30 min before radiation, was ineffective to improve survival of SMMC-7221 and BEL-7402 cells compared with irradiation treatment alone. In contrast, the exposure to SOR, 24 h after irradiation, increased the sensitivity of irradiated cells significantly in a time-dependent manner [80]. Huang et al. [81] showed resistance to radiation-induced apoptosis on four HCC cell lines (PLC5, Huh-7, Sk-Hep1, and Hep3B). However, the presence of SOR 24 h after irradiation reversed this resistance in vitro and in vivo via downregulating phosphorylation of Signal Transducer and Activator of Transcription 3 (STAT3). A similar sequential effect has been observed by Wild et al. in three human (HepG2, HuH7, and Hep3B) and one murine (HCC-4-4) cell lines. Concurrent treatment provided radioprotection in three of four cell lines, whereas sequential treatment yielded lower survival for all four cell lines through apoptosis compared to radiation alone. Radioprotection was mediated by cell cycle slowing and reassortment through G1-S delay. The concurrent treatment induced more repair of DNA double-strand breaks (DSBs) than RT alone. In vivo, sequential treatment compared with concurrent treatment or RT alone produced the greatest tumor growth delay, the most persistent DSBs, the greatest antiangiogenic effect, and the greatest reduction of mitotic index [82].

Nevertheless, the sequential effect favored SOR pre-radiotherapy in other reports. Dai et al. have shown that SOR administered 1 h before RT resulted in radiosensitization of HCC cell line SMMC-7721 [83]. Chen et al. found that tumor growth inhibition was the best in the pre-RT group compared with concurrent or post-RT in three human HCC cell lines [84]. Pretreatment with SOR suppressed most significantly the radiation-induced expressions of nuclear factor kB (NF-kB) and its downstream proteins through suppression of radiation-activated phosphorylation of extracellular signal-regulated kinase (pERK).

Finally, other authors have found that SOR increased the radiation effects on human HCC cells when administered pre-treatment, concurrently, and post-treatment, through different mechanisms and in different amount depending on cell lines [8].

In summary, preclinical data suggest that SOR may act as a radiosensitizer in HCC cell lines, even though the most convenient treatment sequence has not been dilucidated in the laboratories.

#### 4.2.2. Clinical Studies

The combination of SOR and RT is particularly interesting in bad prognosis circumstances such as PVTT, where SOR monotherapy yields very poor survival. The efficacy of conventionally fractionated RT (once per day at 2.0 to 2.5 Gy per fraction up to 50 Gy) with concurrent and sequential SOR therapy (400 mg twice a day) was evaluated in a phase 2 study in 40 patients, 24 with PVTT, with unresectable HCC unfit for TACE. The initial objective response rate was 55% with a 2-year infield PFS of 39% [85]. However, there was a significant rate of severe toxicity, as well as three grade 5 events. These strategy results were not particularly superior to the results from some RT alone retrospective studies [86].

Concurrent SOR and SBRT (six fractions) was tested by Dawson et al. in a phase 1 study for advanced HCC on 16 CTP Class A patients stratified according the effective irradiated liver volume (Veff). Sorafenib was delivered for 12 weeks with SBRT (median dose 51 Gy, range 39–54 Gy) administered during weeks 2 and 3. The study found that for small tumors (Veff <30%), concurrent use of SBRT and SOR 400 mg po daily appeared tolerable with no dose limiting toxicities. However, in high tumor volumes (Veff 30–60%), 200 mg daily was the maximally tolerated dose [87,88].

Que et al. [89] retrospectively analyzed 54 HCC patients with PVTT treated with SBRT (36–45 Gy in 3–5 fractions), where 18 patients received combined treatment with SOR (400 mg twice a day). The combined treatment group demonstrated superiority in median survival (12.5 months vs. 7 months), 1 and 2-year OS rate (55.6% vs. 33.3% and 17.7% vs. 11.1%), and 1 and 2-year PFS (25% vs. 11.1% and 15.2% vs. 8.3%), but these differences were not statistically significant.

Unexpected liver volume reductions were observed in patients with focal liver lesions after 7 days of SOR prior to liver SBRT, but not in patients treated with SBRT alone, suggesting an effect of SOR on irradiated liver [90]. This observation adds complexity to the already difficult radiotherapy planning process of liver tumors.

Therefore, concurrent SBRT with SOR is still experimental treatment and should not be recommended outside clinical trials. The use of SBRT followed by SOR is currently under evaluation in patients with BCLC B or C, with or without vascular invasion, unsuitable for surgical or invasive procedures in a phase III trial (ClinicalTrials.gov Identifier: NCT01730937) by the Radiation Therapy Oncology Group (RTOG 1112).

### 4.3. Antiangiogenics

The antiagiogenic drug vandetanib is a TKI of vascular endothelial growth factor receptor-2 (VEGFR-2) and RET proto-oncogene with radio-enhancement potential. Znati et al. [91] studied outcomes following combined treatment in pre-clinical models. In addition to conventional in vitro assays, the combination was studied in 3D spheroids and in a syngeneic mouse model of HCC. The combined treatment reduced cell survival and inhibited both cell migration and invasion in vitro. In vivo, combination therapy significantly increased tumor growth delay and improved survival.

Apatinib is a TKI highly selective for VEGFR-2. Apatinib efficacy is not high, necessitating association with other therapies (chemotherapy or immunotherapy) for better results. It has been demonstrated that the radiosensitivity of HCC cell lines is increased by apatinib through repair of radiation-induced DNA DSBs suppression and increased radiation-induced apoptosis mediated by suppression of radiation-induced phosphatidylinositol-3-kinase (PI3K)/AKT pathway. The potentiation effect was observed also in vivo, where combined treatment decreased xenograft tumor growth [92]. A Chinese phase III trial (ChiCTR1900027102) will assess the efficacy and safety of SBRT combined with apatinib (anti-PD1 ICI) and camrelizumab for HCC patients with PVTT [93].

### 4.4. Mammalian Target of Rapamycin Inhibitors

The PI3K/protein kinase B (PKB, AKT)/mammalian target of Rapamycin (mTOR) signaling is implicated in the acquisition of a radiation resistant phenotype by cancerous cells of different histological origin [94]. Constituents of the PI3K/AKT/mTOR signaling cascade could be a target for radiosensitization strategies [94,95,96]. Overactivated mammalian target of rapamycin (mTOR) plays an important part in the resistance of HCC to radiotherapy. Thus, mTOR inhibitors have potential as novel radiosensitizers to enhance the efficacy of RT for HCC. A new class of small-molecule selective mTOR inhibitors has been developed and their radiosensitization effects on HCC cells was observed in vitro [97].

### 4.5. Sonic Hedgehog Inhibitors

Sonic Hedgehog (SHH) is a regulator in HCC tumorigenesis [98]. Activation of SHH signaling protects HCC cells against ionizing radiation [99]. Treatment with irradiation and the steroidal alkaloid cyclopamine, an SHH inhibitor, exhibited higher antiproliferative potential than either modality alone, probably due to apoptosis induction. The combined treatment reduced the mean tumor size of orthotopic tumors when compared with radiotherapy alone [100].

### 4.6. DNA Repair Inhibitors

There are different types of strategies to radiosensitize HCC cells through inhibition of DNA repair process. Herein, we describe some recent preclinical studies involving DNA repair inhibitors in combination with RT.

For DNA DSBs, there are two main repair pathways, homologous recombination (HR) and non-homologous end-joining (NHEJ). The activity of DNA-dependent protein kinase (DNA-PK) is essential for NHEJ pathway [101]. As NHEJ occurs throughout the entire cell cycle and can complete the repair process rapidly, targeting of DNA-PK catalytic activity is a plausible strategy for radiosensitization [101]. This rationale is particularly appropriate for liver tumors, because DNA-PK activation and gene expression have been shown to be elevated in HCC, and correlate with a poor response to standard therapies [102]. Presently, it is not clear whether transient pharmacological inhibition of DNA-PKcs can be applied jointly with radiation without increasing toxicity at the same time. Willoughby et al. [103] reported the identification of NU5455, a novel highly selective orally bioavailable inhibitor of DNA-PKcs activity. NU5455 was used to evaluate pharmacological DNA-PKcs inhibition in combination with RT in a cell line panel, including HCC cell lines, by examining the responses of both tumor and normal tissues to treatment. The results indicated that selective and transient DNA-PKcs inhibition in vivo can increase an RT antitumor response without significantly affecting DNA repair in normal tissues and, therefore, without potentiation of acute or late toxicity.

Radiosensitization with drugs already tested in clinical trials with clinical approved indications is particularly interesting. Palbociclib is an oral cyclin-dependent kinase (CDK) inhibitor tested currently, alone or in combination strategies, for various types of malignancies in numerous active clinical trials. Huang et al. evaluated experimentally palbociclib in combination with RT for treating two human HCC cell lines and one cholangiocarcinoma (CCA) cell line and addressed the molecular mechanism behind the combination therapy. Palbociclib delayed the DNA repair process and enhanced the radiation sensitivity of HCC and CCA cells. Importantly, they found that palbociclib inhibited ataxia telangiectasia-mutated (ATM) kinase, the key DSBs sensor involved in HR repair pathway, explaining the synergism between palbociclib and RT [104].

Other DNA repair inhibitors in clinical use that show potential as radiosensitizers in HCC include poly(ADP-ribose) polymerase (PARP) inhibitors [105,106].

### 4.7. Nanoparticles

In addition to drugs with radiosensitization properties, cancer nanotechnology has received considerable attention recently for the treatment of HCC. Nanoparticles (NPs) can increase radiation sensitivity of tumor tissue through their physical properties or through their loaded drugs biochemical properties. For instance, NPs with high Z atoms, such as gold (Gold NPs or GNPs), may induce strong radiosensitization on tumors. Gold NPs have attracted much attention, owing to their biocompatibility and their superior optical properties [107]. Gold NPs enhanced radiosensitivity on HCC cells, increasing DNA damage and apoptosis [108,109]. Hybrid nanoparticles are able to combine imaging and radiotherapy enhancement in order to perform highly reliable cancer theragnosis [109]. Gadolinium NPs (GdNPs), with a high number of gadolinium atoms (*Z* = 64), can be used for enhanced magnetic resonance imaging (MRI), as well as radiosensitization. Even passively absorbed, their liver uptake is much lower than that in HCC. These materials have good biosafety and biocompatibility at conventional therapeutic concentrations. In vitro evaluation demonstrated that the presence of GdNPs significantly decreased HepG2 cell survival when combined with an X-rays in a dose-dependent manner [110].

The NPs are often modified with an agent able to interact with a target molecule on the tumor cell surface to promote a specific organ or tissue delivery of drugs, increasing nanoparticle specificity for cell binding and uptake. This is called active targeting. For instance, Cetuximab (C225) can act as a targeting agent for the extracellular domain of epidermal growth factor receptor (EGFR). Cetuximab attached to GNP surfaces (C225-GNPs) increased the GNP targeting specificity being effectively uptaken by SMCC7721 HCC cells and markedly enhancing cancer cell death through apoptosis [111]. The C225-GNPs significantly increased the radiation-induced suppression of tumor growth in orthotopic SMCC7721 xenografts, acting as potent radiosensitizers for HCC with the EGFR overexpression.

Asialoglycoprotein receptor (ASGPR) is a hepatocyte-specific receptor abundantly expressed on the sinusoidal surface of liver cells. ASGPR mediates the capture and endocytosis of galactose- or N-acetylgalactosamine-terminating glycoproteins. When galactose is coupled to GNPs, it can recognize the ASGPR on the HCC, which improves GNPs’ ability to bind to HCC and increase radiosensitization [112]. However, as normal hepatocytes also express ASGPR, there is still a risk of RILD.

About 70% of HCC cell membranes express Glypican-3 (GPC3) molecules serving as target molecule for anti-GPC3 antibody-containing NPs. Dactolisib (BEZ235) is a PI3K/AKT/mTOR pathway inhibitor with anti-tumor potential. However, dactolisib’s lack of specificity can cause important toxicity when administered in vivo. Dactolisib-loaded NPs containing anti-GPC3 achieved a radiosensitization effect improving the RT killing of HepG2 cells [113].

Previous examples are only a glimpse of the nanotechnological approaches for radiosensitization. The field of nanoparticles active targeting is in rapid development with a multitude of potential target molecules on HCC cells surface, such as integrins, transferrin receptor, folate receptor, and glycyrrhetinic acid receptor [114]. New nanoplatforms designed with several functionalities, radiosensitization included, provide a promising strategy for HCC theranostics [115,116].

## 5. Combination of External Radiotherapy and Immunotherapy

As an inflammation-driven cancer, HCC is a potential target for immunotherapy. Discoveries in tumor immunology made in the last decade have raised enthusiasm for combining immune-checkpoint blockade with SBRT to increase the proimmunogenic antitumor effects seen with SBRT alone [117].

### 5.1. Radiotherapy and the Immune Response

Radiation has ambivalent effects on tumor immune microenvironment, promoting and suppressing immunity at the same time. It is well established that RT, particularly SBRT, can promote a potent antitumor immune response through diverse mechanisms. The high doses of radiation used in SBRT increase tumor cell lysis and release a large load of TAA as a consequence. The released TAAs are taken up by professional antigen presenting cells (APC), including dendritic cells and macrophages [118]. Cancer cell death induced by radiation after DNA damage, facilitates double-strand DNA release, activating the cyclic GMP–AMP synthase/stimulator of interferon genes (cGAS/STING) signaling pathway; as a consequence, innate immunity is stimulated and lymphocyte infiltration is increased in cancer tissues. Proinflammatory cytokines can then activate the APCs that migrate in turn to tumor-draining lymph nodes. Here, CD8+ cytotoxic T cells are activated to provide antitumor immunity [119]. In addition to enabling the mobilization of T cells against cancerous cells, radiation results in the translocation of calreticulin to the tumor-cell surface [120]. This serves as a signal to activate macrophages and DCs, which internalize calreticulin-expressing tumor cells.

Although RT alone contributes to the conditions necessary for the cross-priming of cytotoxic T lymphocytes against TAAs, the effect of RT on immunologic tumor microenvironment appears to be relatively weak or even null, showing most commonly immunosuppressive net results which favor a tumor escape from immune responses and facilitates tumor recurrence. The mechanism for the immunosuppressive effect involves also the activation of DNA damage response pathway started by DNA damage that stimulates the expression of CTLA-4 and programmed death ligand 1 (PD-L1) on the tumor cell membrane [121,122]. The junction of PD-L1, with its receptor PD-1 in T cell membranes, induces T cell exhaustion and results in tumor escape from the host immune response.

With the goal to suppress the brake for the anti-tumor immune response created by the PDL1/PD-1 interaction, immune checkpoint inhibitors (ICIs) administered jointly with RT could have a synergistic effect, tipping the balance in favor of a proimmunogenic tumor microenvironment [122].

### 5.2. Preclinical Studies in HCC Models

Preclinical studies and mechanisms involved in radiation and immunotherapy synergistic effect have been reviewed in recent publications [118,123,124]. At least three preclinical studies with RT and ICIs have been published using murine HCC models. Kim et al. showed a positive effect when anti-PD-L1 and RT (10 Gy) were combined, delaying Hca-1 tumor growth in syngeneic C3H mice and improving the survival of tumor-bearing mice compared to those with one of the treatments alone [125]. Tumors exposed to the combined treatment exhibited higher infiltration of effector CD8+ T cells and more apoptosis within tumor tissue after RT. Friedman et al. corroborated the synergistic effect of SBRT (30 Gy in three fractions) and anti-PD-1 combination in an orthotopic murine HCC model resulting in tumor growth suppression, longer survival, and more profuse infiltration of CD8+ cytotoxic T cells within the tumor [126]. Sheng et al. tried in the laboratory a trimodality concomitant therapy: radiotherapy (3 × 6 Gy), anti-PD-L1, and the DNA repair inhibitor AZD6738. Tumor immune microenvironment was significantly changed with AZD6738 treatment. The triple therapy displayed better results in antitumor efficacy and mice survival than RT plus anti-PD-L1. Mechanistically, the abrogation of DNA repair by AZD6738 rendered a more potent activation of cGAS/STING signaling pathway by RT, favoring the synergism between RT plus anti-PD-L1. Furthermore, triple therapy led to stronger immunologic memory and lasting antitumor immunity than RT plus anti-PD-L1, thus preventing tumor recurrence [127].

These findings indicate that immune activation of tumor microenvironment with ICIs alone or mixed with other radiopotentiators might be a powerful synergistic treatment for SBRT on HCC patients.

### 5.3. Clinical Studies

#### 5.3.1. Immune Checkpoint Inhibitors for HCC

After decades of research, the hope of effective immunotherapy for HCC patients became a reality with the development of ICIs. The landmark study, the phase 1/2 multi-cohort dose escalation and expansion trial CheckMate 040, showed that nivolumab (3 mg/kg), an anti-PD-1 antibody, had a manageable safety profile and durable objective responses (20%) with a median OS of 15 months [128]. The following phase III study CheckMate-459 showed a marginal superiority of nivolumab over sorafenib in improving OS as a first-line therapy for unresectable HCC [129]. The efficacy of another anti-PD-1 monoclonal antibody, pembrolizumab, was assessed in a phase II open-label trial (KEYNOTE-224) obtaining an objective response rate of 17% with good tolerance in SOR-experienced patients [130]. Nivolumab and pembrolizumab received accelerated approval in patients with advanced HCC in the second-line setting based on these results. The phase 3 KEYNOTE-240 study followed these early studies to further assess the utility and safety of pembrolizumab in patients with advanced HCC after failing first-line treatment. Despite the lack of significant survival benefit, pembrolizumab did show a response rate of 17% with a median duration of response of 13.8 months [131].

Several established and novel ICIs are being evaluated for use in advanced setting, first-line and after failed systemic therapies. Since the response rate of ICIs is only 15–20%, efforts to find an optimal combination with other treatments are underway to further increase the therapeutic effect. The strategies followed at present time include dual immune checkpoint blockade, combination of ICIs with targeted therapies, mainly TKIs and antiangiogenics, and a combination with local therapies, such as TACE, TARE, and SBRT [132,133].

#### 5.3.2. Clinical Studies with SBRT and ICIs in HCC

Mounting evidence suggests local radiotherapy might promote immunogenic tumor cell death, and ICIs could favor a stronger immunologic anti-tumor reaction after treatment [118,119,120,121,122,123,124,125,126,127]. Despite this, there is no published prospective clinical data on combined RT and ICIs therapy in HCC, except a few small series [134].

Several ongoing clinical trials are being conducted to investigate the efficacy of RT using various ICIs and external-beam RT for patients with HCC (Table 7).

NCT03482102 is a phase II trial to test the safety and effectiveness of durvalumab, tremelimumab, and hypofractionated RT in HCC and biliary tract cancer patients [135]. NCT03203304 is a phase I trial of SBRT (40 Gy in five fractions) followed by nivolumab every two weeks as monotherapy or in combination with ipilimumab every 6 weeks in unresectable HCC [136]. NCT03316872 is a phase 2 study to assess the efficacy of the combination of pembrolizumab and SBRT in patients with advanced HCC who have experienced disease progression after treatment with SOR [137]. NCT03817736 is a prospective phase II single arm study, assessing the efficacy and safety of the sequential administration of TACE and SBRT with an ICI in HCC patients for downstaging before hepatectomy [138].

The probability of achieving clinical benefit from ICIs therapy is restricted to a fraction of patients with HCC. There is an urgent need to find predictive factors of response and survival to spare patients from potentially ineffective and toxic therapies. In this sense, the study of inflammatory biomarkers after RT becomes particularly interesting. Kim et al. [139] found that the soluble PD-L1 (sPD-L1) level was clinically relevant in blood samples from 53 patients with HCC treated with conventional RT or SBRT. The initial sPD-L1 level was significantly associated with advanced stage features such as tumor size or PVTT, and was a significant prognostic factor inversely correlated with OS. At one month post-treatment, sPD-L1 was significantly related to early lung metastasis. The sPD-L1 level was significantly increased after RT and the change pattern of sPD-L1 was different between the two RT schemes: sPD-L1 tended to increase continuously until 1 month in SBRT patients but decreased in conventional RT patients. This suggests that SBRT might be better than conventional fractionated RT for combined use with ICIs.

## 6. Conclusions

Recent developments in image guidance and radiation delivery technology have been crucial for safe administration of high-dose external beam RT. The application of SBRT in the treatment of HCC is changing the status of RT, increasing its consideration in the multidisciplinary treatment protocols. Both prospective and retrospective studies demonstrate high rates of local control with limited hepatic and gastrointestinal toxicities, but the intrahepatic recurrence rates outside the irradiated field remain high. Based on available evidence, SBRT could be a potential alternative therapy for small HCC, especially if the tumor is unresectable or not amenable for consolidated ablative treatments, and for advanced tumors in combination with TACE. New SBRT developments, such as Magnetic Resonance guided treatments or proton-therapy, have been already introduced in the clinic, showing decreased liver dose with a potential for decreased toxicities.

As systemic therapies improve, loco-regional therapies become more relevant. Multiple clinical trials utilizing SBRT in combination with other treatment modalities, which include systemic or local therapies, are underway. Particularly interesting are the radiosensitization strategies with nanoparticles, PKIs, and ICIs. Little is known yet about immunotherapy’s role in resectable disease, following locoregional therapy or combined with SBRT for unresectable tumors.

The role of SBRT will not be recognized in a multi-professional setting without a proper amount of evidence. Phase III trials and predictive biomarkers research are eagerly needed to further establish the status of SBRT, alone or potentiated with other treatments, for HCC patients.

## Figures and Tables

**Table 1 cancers-13-00192-t001:** Hepatocellular carcinoma radiosensitivity described in terms of SF2 and LQ parameters.

Cell Line	Parameter	Number of Cell Lines	Mean	Median	Min.	Max.	SD	References
All tumors excluding HCC	SF_2_	134	0.37	0.35	0.001	0.86	0.20	[8]
HCC (primary cultured cells)	SF_2_	29	0.41		0.28	0.78	0.05	[7,9]
HepG2, Hep3b	SF2	2			0.34	0.67		[5]
	α/β (Gy)				3.1	7.4		
	α (Gy^−1^)				0.118	0.413		
	β (Gy^−2^)				0.038	0.056		
HepG2, Hep3b	α (Gy^−1^)	2			0.185	0.249		[6]
	β (Gy^−2^)				0.124	0.172		
SMMC-7721, SK-HEP-1	α /β (Gy)	2			1.57	6.64		[8]
	α (Gy^−1^)				0.09	0.37		
	β (Gy^−2^)				0.05	0.06		

HCC, hepatocellular carcinoma; SD, standard deviation.

**Table 2 cancers-13-00192-t002:** Initial dose finding trials of SBRT for HCC.

Study	Year	Sample Size	Dose Schedule	Median Follow-Up (Months)	OS Rate	LC	Toxicity
Méndez Romero [15]	2006	8 (HCC)	37.5/3 * 30/3 25/5	12.9	75% 1 year 40% 2 years	75%	2 RILD 1 Grade 5
Tse [16]	2008	31 (HCC)	36/6 median	17.6	48% 1 year	65%	No grade ≥ 4
Andolino and Cárdenes [17]	2011	60	48/3 40/5	27	67% 2 years	90%	No grade ≥ 3
Bujold [18]	2013	102	36/6 median	31.4	55% 1 year 34% 2 years	87%	7 Grade 5

* 37.5 Gy/3 fractions; OS: Overall Survival; LC: Local Control; RILD: Radiation-Induced Liver Disease.

**Table 3 cancers-13-00192-t003:** Results of SBRT for HCC (small tumors) in the largest retrospective studies and systematic reviews.

Author	Study Design	Number of Patients	Lesion Size	Median Follow-Up (Months)	LC After 3 Years (%)	3 Year OS (%)	Relevant Toxicity
Sanuki [19]	Retrospective	185	GTV 7.64 mL	24	91	70	2 pts. Grade 5
Mathew [20]	Retrospective	297	42% > 3 cm		86.7	39	Worsening of C–P score by ≥2 points in 15.9%
Dobrzycka [21]	Systematic review	973	Average diameter 23 mm	11.5–41.7	93	73.4	5.3% Grade ≥3
Rim [22]	Systematic review	1950	Average diameter 33 mm		83.9	48.3	4.7% Grade ≥3

**Table 4 cancers-13-00192-t004:** SBRT versus hepatectomy: retrospective studies.

Author	Arm	Number of Patients	Survival after 5 Years (%) (after PSM)	*p* (after PSM)	Complications
Su [24]	SBRT	82	70.0 (74.3)	0.558 (0.932)	Fewer hepatic hemorrhage, hepatic pain, and weight loss.
	Resection	35	64.4 (69.2)		Fewer nausea.
Nakano [25]	SBRT	27	(47.8)	0.0005	Grade ≥ 3, 9.1%
	Resection	254	(75.2)		Grade ≥ 3, 3.7%
Sun [26]	SBRT	122	(71.0)	(0.673)	Grade ≥ 3, 0%
	Resection	195	(70.7)		Grade ≥ 3, 21.54%

PSM: Propensity Score Matching.

**Table 5 cancers-13-00192-t005:** SBRT versus Percutaneous Tumor Ablation (PTA): Multicentric retrospective studies and metanalysis.

Author	Arm	Number of Patients	Tumor Control (%)	*p* (After PSM)	Complications Grade ≥ 3 (%)
Kim [39,40]	SBRT	496	3-year LR 21.2	<0.001 (<0.001)	1.6
	PTA	1568	3-year LR 27.9		2.6
Pan [41]	SBRT	859	-	0.003	-
	PTA	1873	-		-
Lee [42]	SBRT	779	2-year LC 84.5	0.431	0–11.4
	PTA	1211	2-year LC 79.5		0–6.4

LR: Local Recurrence; LC; Local Control; PSM: Propensity Score Matching.

**Table 6 cancers-13-00192-t006:** SBRT versus TACE: Retrospective studies.

Author	Arm	Number of Patients	Local Control 2–3 Years (%)	*p* (after PSM)	Survival 2–3 Years (%) (after PSM)	*p* (after PSM)	Complications (%)	*p*
Sapir [46]	SBRT	125	91.3	(<0.001)	34.9	(0.21)	Grade ≥ 3, 13	0.05
	TACE	84	22.9		54.9		Grade ≥ 3, 8	
Shen [47]	SBRT	46	73.3	(0.007)	47.4	(<0.001)	-	-
	TACE	142	63		22.9		-	
Su [48]	SBRT	167	62.5	(0.0047)	(65.1)	(0.29)	Grade 5, 1.2CP + 2 score, 10.1	0.11
	TACE	159	53.3		(61)		Grade 5, 1.3CP + 2 score, 4.7	

CP + 2 score: Elevated Child–Pugh score at two points.

**Table 7 cancers-13-00192-t007:** Ongoing clinical trials on external-beam RT and Immune Checkpoint Inhibitors for patients with HCC.

NCT Number	Institution	Phase	Disease	Intervention: ICI and Local Treatment		Estimated Enrollment
03482102	MGH	II	Locally advanced/unresectable or metastatic diseaseHCC or biliary tract cancer	Tremelimumab + Durvalumab	30 Gy/3 fx	70
03203304	UCh	I	HCC unresectable	Nivolumab	SBRT 40 Gy/5 fx	50
03316872	UHN	II	HCC showing progression after sorafenib	Pembrolizumab	SBRT 5 fx	30
03817736	UHK	II	HCC prior hepatectomy	ICI	TACE + SBRT	33

MGH: Massachusetts General Hospital; ICI: Immune Checkpoint Inhibitor; SBRT: Stereotactic Body Radiation Therapy; TACE: Trans-Arterial Chemo-Embolization; UCh: University of Chicago; UHN: University Health Network (Toronto); UHK: University of Hong Kong.

## Data Availability

Not applicable.

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
