# Peer review of "Stereotactic Radiotherapy for Hepatocellular Carcinoma, Radiosensitization Strategies and Radiation-Immunotherapy Combination"

_cancers, 2021, doi:10.3390/cancers13020192_

Round 1

Reviewer 1 Report

Dear Colleagues,

The paper summarises the current evidence in a very structured way and compares the oncological effectiveness with other therapeutic procedures - alone or in combination. The work is very comprehensive and, despite its size, very carefully researched and sensibly structured. For this reason alone, the work should be published. Beyond that, however, it conveys valuable content that can advertise more understanding of the interrelationships and serve as a basis for further interdisciplinary research.

The work does not need any fundamental change, only the following careless mistakes should be revised.

Spelling mistakes

236 – 272 – either PTA or TPA

445 – „improve“ instead of „reduce“ would make more sense

Best

Author Response

Spelling mistakes

236 – 272 – either PTA or TPA

Percutaneous Tumor Ablation has been abreviated by PTA; TPA has been removed.

445 – „improve“ instead of „reduce“ would make more sense

"reduce" replaced by "improve"

Reviewer 2 Report

The paper entitled STEREOTACTIC RADIOTHERAPY FOR HEPATOCELLULAR CARCINOMA, RADIOSENSITIZATION STRATEGIES AND RADIATION-IMMUNOTHERAPY COMBINATION presents the current state-of-the-art of treating  patients diagnosed with non-operable hepatocellular carcinoma with a highly selective and precise tool, Stereotactic Body Radiotherapy (SBRT). 

The concept of radiotherapy utilization in hepatocellular carcinoma treatment is first introduced, and then the Authors concentrate on SBRT alone or in combination with other treatment modalities, like percutaneous tumor ablation, transartelial chemoembolization, transartelial radioembolization, chemotherapy, and immunotherapy. 

This is a well-prepared and written paper providing sound and interesting information on the topic. 

The only problematic issue I found relates to the inconsistent use of the abbreviation for Percutaneous Tumor Ablation; both PTA and TPA are used without providing any justification. 

Author Response

The only problematic issue I found relates to the inconsistent use of the abbreviation for Percutaneous Tumor Ablation; both PTA and TPA are used without providing any justification.

The inconsitancy has been solved adopting PTA as the appropriate abbreviation for Percutaneous Tumor Ablation across the manuscript.

Reviewer 3 Report

Comprehensive review about SBRT of HCC.

I have only minor comments:

-Please add a table/figure summarizing all the main results of each paragraph

-Please add some data about MR-guided SBRT or any high-end technologies in that field

Author Response

-Please add a table/figure summarizing all the main results of each paragraph

Five tables have been added in those sections where the studies were considered homogeneous enough to provide meaningful comparisons.

-Please add some data about MR-guided SBRT or any high-end technologies in that field

A review extension into "high-end technologies", as suggested by the reviewer, will increase the size of an already oversized manuscript because, not only MR-guided SBRT, but also protontherapy, are the new technological advances for SBRT, and both should be included. Furthermore, technological aspects of SBRT have been intentionally omitted since the manuscript target population is multidisciplinary. A mention to new RT technological developments have been added in the manuscript conclusion but not providing more data or references:

"New SBRT developments like Magnetic Resonance guided treatments or proton-therapy have been already introduced in the clinic showing decreased liver dose with a potential for decreased toxicities"